# Exploring the Interpad Gap Region in Ultra-Fast Silicon Detectors: Insights into Isolation Structure and Electric Field Effects on Charge Multiplication

**DOI:** 10.3390/s23156746

**Published:** 2023-07-28

**Authors:** Gordana Laštovička-Medin, Mateusz Rebarz, Jovana Doknic, Ivona Bozovic, Gregor Kramberger, Tomáš Laštovička, Jakob Andreasson

**Affiliations:** 1Faculty of Natural Sciences and Mathematics, University of Montenegro, Dzordza Vashingtona, 81000 Podgorica, Montenegro; 2ELI Beamlines Facility, The Extreme Light Infrastructure ERIC, Za Radnicí 835, 25241 Dolní Břežany, Czech Republic; mateusz.rebarz@eli-beams.eu (M.R.); jakob.andreasson@eli-beams.eu (J.A.); 3Jozef Stefan Institute, Jamova Cesta 39, 10 000 Ljubljana, Slovenia; gregor.kramberger@ijs.si; 4Institute of Physics, Academy of Sciences of the Czech Republic, Na Slovance 2, 18221 Prague 8, Czech Republic; tomas.lastovicka@eli-beams.eu

**Keywords:** segmented UFSD, isolation structure, interpixel region, fs-laser, TCT

## Abstract

We present an in-depth investigation of the interpad (IP) gap region in the ultra-fast silicon detector (UFSD) Type 10, utilizing a femtosecond laser beam and the transient current technique (TCT) as probing instruments. The sensor, fabricated in the trench-isolated TI-LGAD RD50 production batch at the FBK Foundry, enables a direct comparison between TI-LGAD and standard UFSD structures. This research aims to elucidate the isolation structure in the IP region and measure the IP distance between pads, comparing it to the nominal value provided by the vendor. Our findings reveal an unexpectedly strong signal induced near p-stops. This effect is amplified with increasing laser power, suggesting significant avalanche multiplication, and is also observed at moderate laser intensity and high HV bias. This investigation contributes valuable insights into the IP region’s isolation structure and electric field effects on charge collection, providing critical data for the development of advanced sensor technology for the Compact Muon Selenoid (CMS) experiment and other high-precision applications.

## 1. Introduction

Low Gain Avalanche Detectors (LGAD) are silicon detectors based on the avalanche process initiated by a charge moving in a high electrical field with a low internal gain (O(10) [1]). The internal gain layer is a doped region of the same sign as a substrate (a layer of acceptors with appropriate charge density, ρ_A_ ~ 10^16^ cm^−3^, ensuring the electric field E ~ 300 kV/cm needed for impact ionization and charge multiplication). This layer is implemented close to the np junction. Consequently, larger signals are measured in the LGADs compared to PIN diodes; the low gain limits the shot noise and optimizes the signal-to-noise ratio (SNR). An important aspect of LGAD design is its thickness. It was demonstrated that thinning the active width of LGADs from 300 μm [2] to 50 μm [3] improved the timing resolution of the sensor. The active sensor width was further reduced down to 35 μm and 24 μm [4]. Fast and thin LGAD sensors are now called ultra-fast silicon detectors (UFSD) [5] and are fabricated at the Fondazione Bruno Kessler (FBK). Thanks to the moderate internal gain (between 10 and 70), SNR is improved; for a 45 µm thick UFSD pixel and gain of 20–30 times resolution of ~30 ps [6] in a beam test setup has been achieved. A detailed description of the UFSD characteristics can be found elsewhere [7,8]. The key points of LGADs optimized for timing are as follows: large and fast enough signal to assure excellent timing performance while maintaining almost unchanged noise levels (low jitter term), reduced Landau fluctuations (≈50 µm thin sensors), and a very uniform weighting field. The latter has an important impact on the timing performance of the device because the non-uniformity of the weighting field brings the distortion of the signal [9,10]. As pointed out in ref [9], in every particle detector, the shape of the induced current signal can be calculated using Ramo’s theorem [10] that states that the current induced by a charge carrier is proportional to its electric charge q, the drift velocity v, and the weighting field E_w_: i(t)~qvE_w_. This relationship indicates the two key points in the design of sensors for accurate timing. First, the drift velocity needs to be constant throughout the volume of the sensor, otherwise the non-uniform drift velocities induce variations in signal shape as a function of the hit position. As a result, the overall time resolution is spoiled.

A timing resolution of 30 ps achieved for minimum ionizing particles (MIP) and the ability to tolerate radiation hardness up to 2.5 × 10^15^ n_eq_/cm^2^ (1 MeV neutrons equivalent flux, used as a measure for displacement damage dose), which is expected during the high luminosity of the Large Hadron Collider (HL-LHC), made the UFSD a very attractive sensor technology not only in high energy physics (HEP) experiments but also for medical applications [11,12,13].

Additionally, UFSDs introduce the concept of four-dimensional (4D) tracking [14] of charged particles, which requires simultaneous spatial resolution in the range of O(10) µm and time resolution in the range of O(10) ps. However, achieving such high resolutions using standard segmented UFSD technology remains a challenge due to the difficulty in achieving uniform gain layer doping over a large area. Segmented LGADs face the challenge of a high electric field at the junction periphery, which is addressed through the inclusion of a p-doped implant (p-stop) and an n-type region called junction termination extension (JTE) in the border region. These components provide isolation and control of the junction curvature, reducing the electric field at the pads periphery and preventing unwanted ionization multiplication. However, this can lead to a decreased gain region and reduced fill factor.

Obviously, the technological and physical limitations define the lower boundary of the no-gain region in segmented LGADs, with the minimum achieved distance being 38 μm in FBK UFSD sensors. The size of the interpad no-gain area impacts the array fill factor, with a smaller no-gain area resulting in a higher fill factor. Aggressive designs (with very small distance between neighbour pixels) and alternative approaches, such as trench-isolated LGAD (TI-LGAD) [15] and resistive AC-coupled LGAD (RSD) projects [16], aim to maximize the fill factor of the sensor array.

In this study, we present the space–charge interpad region characterization of the prototypic segmented UFSD sensor (Type 10, FBK production). The collected charge profiles and individual transient current waveforms, measured under different bias and illumination conditions, are presented and discussed.

## 2. Materials and Methods

### 2.1. UFSD Productions Description

Over the past seven years, FBK has conducted research and development on various UFSD productions. Each version of UFSD introduces certain features and improvements. The main motivations behind each UFSD production are to enhance timing performance, improve charge collection efficiency, increase radiation hardness, and improve the fill factor. The main features and motivations for each of the UFSD productions manufactured by FBK can be summarized as follows. The initial production was performed in 2016 (UFSD1) and involved 300-µm thick LGADs. Then, in 2017, a new UFSD2 production has focused on optimization of the gain layer design of 50-µm thick LGADs. Boron and Gallium were used as gain layer dopants, co-implanted with carbon to improve the radiation hardness. The UFSD3 production from 2018 was the first 50-µm thick LGAD production by FBK. Four Boron doses were combined with four carbon doses for the gain layer, and the three different strategies for gain layer termination structures were designed. The characterization tests showed that UFSD3 production was partially affected by early breakdown of some prototype devices and by a very high random noise; the combination of aggressive pad termination designs and incorrect p-stop doping concentration were found as reasons. This was solved through the dedicated FBK production run, and the correct p-stop dose range was defined. In 2019, UFSD 3.1 prototypes were produced, followed by UFSD 3.2 in 2020. UFSD 3.1 run has been focused on studying the interpad region and on the optimization of the p-stop dose; for this purpose, the eleven different layouts for gain layer termination structures were designed and implemented. They were further tuned in the UFSD 3.2 production; the latter was dedicated to the key optimization requirements such as: (1) the investigation of a lower carbon dose to be co-implemented with boron in the gain layer aiming to improve the radiation hardness; (2) in addition to the shallow gain layer, the deep gain layers have been explored and two different depths of the gain layer were explored with the aim of improving operating parameters of highly irradiated devices; (3) to better understand the aggressive design (its limits and drawbacks) in order to reduce the no-gain region and thus to increase the fill factor of segmented LGADs that is requested for a good spatial resolution and consequently, for the 4D tracking. The five different doses of boron were combined with four doses of carbon for the gain layer, and different types of diffusion were applied at two depths to optimize the carbon level and utilize deep-carbonated gain implants. In total, in the UFSD 3.2 run, the gain layer termination structures of 19 wafers were fabricated with thicknesses ranging from 45 µm to 55 µm. Finally, UFSD 4.0 production was performed in 2021 for further optimization of the sensor parameters. In this production run, the two IP layouts, Type 9 and Type 10 for HGTD ATLAS and MIP TID CMS, respectively, were further optimized. In the UFSD 4.0 batch run, the Type 9 was produced with a nominal distance of 49 μm and with an interpad isolation structure consisting of a double p-stops while the Type 10 prototype has a nominal IP of 61 μm and the layout of isolation structure is built from two p-stops and bias ring.

### 2.2. Investigated Sensor

In the “RD50 TI-LGAD” batch, several UFSD-like sensors (Type 4 and Type 10) have been produced for comparison purposes. These UFSD-like samples have the same pixel arrangement as the TI-LGAD samples (1 × 2 pixels) and were produced on the same wafers, so they shared the identical manufacturing process flow. One of these prototype sensors, Type 10, was used in this study (Figure 1a). The interpad of this sensor consists of two p-stops and a grid of guard rings (Figure 1b). Therefore, the interpad resistance for Type 10 should be understood as a pad–guard ring resistance.

The implementation of a guard ring and p-stop structures in-between two pads is not a conventional design for pixellated sensors. Here, we explain the reason. There are two ways of isolating the electrodes in p-type sensors: either a common p-stop (both pads share a p-stop) or individual p-stops (each pad has its own p-stop). The latter is a p-stop design that ensures more efficient isolation. The bias ring in-between is a feature used to additionally reduce the peak fields in case of floating pads. It represents the exceptionally safe design on the expense of slightly larger IP distance.

Notably, the investigated UFSD-like prototype, produced in the TI-LGAD RD50 run, differs from the standard UFSD Type 10 LGADs, now accepted as reference CMS UFSD 4.0. The difference is both in terms of the layout and the manufacturing process.

Layout: even if the Type 10 concept is the same (two p-stops and GR between the pixels), the nominal inter-pixel distance is 49 μm, much narrower than 61 μm in the final CMS/ATLAS layout. In addition, both the p-stops’ width and the gaps are reduced in TI-LGAD batches.Manufacturing process: the TI-LGAD process is not the same as the UFSD one in terms of thermal budget, materials, and other aspects. Therefore, the Type 10 samples produced on the TI-LGAD batch are not directly comparable to the standard UFSD Type 10.

Considering the above differences, the UFSD-like Type 10 sensor from TI-LGAD production was chosen to compare UFSD type (p-stops and GR) and trenches-based isolation structures.

### 2.3. Experimental Setup

The space–charge profiles were measured by the transient current technique (TCT). The experimental setup located at the ELI Beamlines facility (Czech Republic) is presented in Figure 1c and was previously described in detail in [17]. The ultrashort NIR laser pulses (800 nm, 50 fs, 1 kHz) were used to generate charge in the investigated sensor. The beam was focused by an objective to a diameter of 1.7 µm (an actual spatial resolution diffraction limit) and generated transient signal was recorded by a 6 GHz oscilloscope. Individual transient current waveforms were recorded at different HV biases and for different pulse energies when the beam was scanned (with 0.5 µm step) across the metallization opening window extending over two neighboring pixels (pads) and the interpad region of the investigated sensor. The space–charge profiles (x-profiles) were constructed by integrating every individual waveform. The pulse energies varied between 0.2 and 5 pJ, whereas the applied HV bias was scanned from 80 to 220 V.

The sensor was mounted in an aluminum housing connected to the T-bias. The output was wire-bonded to both pixels so the oscilloscope measured a cumulative signal of two pixels. In addition, the measured signal was increased by an external amplifier.

## 3. Results and Discussion

### 3.1. Charge Multiplication in the IP Region Observed in Charge–Spatial Profiles

Example x-profiles, representing the charge distribution over the interpad window of the studied sensor, are presented in Figure 2. For the low laser power (pulse energy 1 pJ) and low bias (100 V), an apparent decrease of the charge generation was observed in the interpad region between −25 μm and +25 μm (zero is assigned to the geometric center of the interpad region). This result corresponds very well to the nominal interpad distance (49 μm). Interestingly, the x-profile in the interpad region is not flat, but it has a characteristic hollow shape in the center. This additional signal drop seemed to originate from the n-guard ring located precisely at the center of the interpad gap region in the Type 10 UFSD sensors. From the “hollow” shape and its position, we deduce that the width of the bias ring, interfaced at the center of the IP region, is around 6 µm.

Interestingly, the increase of the pulse energy and/or bias results in the appearance of an additional structure in the interpad area of the x-profile. It is clearly visible in the profile recorded for 5 pJ pulses at 140 V (see Figure 2), where two characteristic bands appear in the positions ± 12.5 μm. This effect is most probably attributed to the presence of two p-stops in the Type 10 sensor design (compare with Figure 1). Based on the characteristic maxima and minima of the x-profiles, we proposed an assignment of the experimental charge–space distribution to the pixel isolation structure presented in Figure 2. It seems that the fs-laser-based TCT setup (see Section 2.3) can precisely resolve the internal structure of the IP region, indicating the positions of individual implants: pad edges, JTE, p-stops, and bias ring.

Moreover, comparing the collected charge for 1 pJ at 100 V and for 5 pJ at 140 V, one can conclude that the latter has a much lower output compared to the expected value (at least > five times, taking into account the ratio of the laser power and the higher bias voltage applied at the higher laser power). This effect is due to the plasma effect, more pronounced at the higher laser power and due to gain suppression induced by large charge density. Large charge density causes screening of the local electric field, disabling the charge to undergo the impact ionisation in the gain layer.

A systematic study of the observed excess in charge collection in x-profiles at different laser power and bias was further performed to gain deeper insight into the observed effects. The results obtained for 1 pJ pulses at different biases are presented in Figure 3a. It was observed that at higher bias, the signal around the central dip significantly increases, forming the characteristic bands located near p-stop positions. The same behavior is visible in Figure 3b when the laser power increases at a constant bias (100 V).

These effects, indicating charge multiplication, are strongly enhanced with a further increase in the laser power and bias when the more pronounced sharp features appear in the profiles (see Figure 3c). Interestingly, simulations of the electric field profiles for a similar sensor also exhibit spikes in the corresponding region [18]. Finally, at a bias > 180 V and pulse energies of 5 pJ, the charge generated in the IP region becomes completely dominant (Figure 3d) and obscures the fine structure visible for lower injection rates. In addition, it was also noticed that at a very high bias voltage, the external signal amplifier saturates, so the absolute scaling of the collected charge cannot be correctly deduced.

The effect of the dramatic increase of the generated charge in the IP region was not observed in the pad area, even though the same conditions (the same charge intensity injection) were applied. A plausible explanation could be that a strong enough avalanche seed seen in the IP region, in proximity to the p-stop, can be sufficient to overcome damping in the plasma. This effect seems to not be initiated in the pad region (under the same initial conditions), perhaps mainly since the charge collection in the pad is considerably faster than in the interpad region. Hence, the temperature effect in plasma and its cooling must play some triggering effect.

### 3.2. IP Distance at Different HV Bias and Pulse Energy

Although the nominal interpad distance is a physical parameter (defined as the length between gain implants), the effective IP distance (from the generated signal point of view) can vary significantly under different conditions. The effective IP distance of the sensor can be understood here as the distance between the main falling edges of the x-profiles. It is clearly visible in Figure 3 that this gap decreases with increasing bias and laser pulse energy. To get a more quantitative insight into this effect, every x-profile was fitted with two error functions (erf) corresponding to the main edges. The IP distance was then determined as the difference between the centers of the corresponding erf functions. The results of this fitting are gathered in Figure 4, presenting IP distance dependence on the bias and laser pulse energy.

By analyzing the data presented in Figure 4, we deduce that the interpad distance decreases with the increased laser power. This can be explained as follows. By increasing the laser power, the induced charge density is increased. The increased charge density, as already explained, leads to the gain suppression in the device. However, the charge collection still increases with increased laser power but not with the rate as it should if the gain would not be suppressed (the local electric field in gain layer is screened by large charge density resulting in reduced impact ionization and thus the reduced charge multiplication). From other side, a strong excess of charge collection is observed in the interpad region which increases with the increased laser power. All over, this leads to an artificially decreased interpad distance compared to the case when the lower laser power is used.

### 3.3. Evolution of Transient Current Signal Generated at Different Positions

In order to learn more about the observed strong charge generation in the IP region, we conducted a study on the shape of the induced current. By comparing the shape of the individual waveforms recorded when the sensor is illuminated by fs-laser at the different positions, one can deduce information about changes in the electric field due to differences in IP region doping (caused by interfacing the structures such as p-stop and bias ring). For this purpose, we selected a few characteristic points based on the previously measured x-profiles. These points are indicated in the insets of Figure 5. They correspond to the pad, JTE, p-stop, and central guard ring. The waveforms were recorded for two different values of applied laser power, 0.2 pJ (Figure 5a) and 5 pJ (Figure 5b), at the bias of 100 V. To better understand the shapes of the waveforms (rising/falling time, broadening), the amplitude normalized signals are also presented in Figure 5c,d.

The first interesting difference is displayed in Figure 5, where the signal for low (0.2 pJ) and high (5 pJ) pulse energy is compared for the pad and the interpad region. In both cases, one can notice that the rising time of the signal becomes faster when the laser moves from the pad to the interpad area (close to JTE and p-stop). This effect is more pronounced at a lower bias than at a higher bias (compare with Figure 6c). The reason is that in the pad region, due to the presence of a gain layer which is a highly doped p-layer (placed 1–2 μm beneath the front n++ electrodes with a very high electric field), the plasma-like cloud of e-h carriers is created at much lower bias than is required in the IP region. Consequently, the impact of the electric field screening (lowering the drift velocity of electrons) happens faster than in the IP region at the same applied laser power and HV bias. Conversely, in the IP region, the plasma needs more time to build at the same applied bias. The plasma formation goes slower in the interpad region since the initial value of the electric field here is lower. Consequently, the drift velocity of the electron is higher since the damping caused by the plasma is less effective.

On the other hand, when the laser illuminates the central GR, the rising time of the signal becomes slower again and is comparable with the pad area. This is expected since the diffusion effect at a low field (where the bias ring is interfaced) is larger. Furthermore, the drift time is longer since there is a longer drift path for the charge to travel the distance from the position where it is generated to the position of JTE where it is collected (slow drift due to the low doped region).

The apparent plasma damping in charge multiplication can be deduced by comparing the amplitudes. For instance, the pad signal recorded for 0.2 pJ exhibits an amplitude of about 9 mV, whereas for 25 times higher power (5 pJ) it is only about 5.5 times higher (amplitude of about 50 mV).

The pattern of the plasma formation behavior and its erosion in low and highly doped regions is presented in Figure 6, Figure 7 and Figure 8. The studied prototype sensor was scanned over HV bias, from 80 V to 200 V, while the two different laser energies were chosen to represent the low (0.2 pJ) and the high (5 pJ) charge density injection. The signal from the pad was compared to the signals recorded at the locations where the p-stops and bias grid were interfaced.

The amplitude of the pad signal increases clearly with the bias increase, which indicates that some gain is attained. At the same time, the rising time of the pad signal becomes faster, and this effect is more pronounced in the case of higher pulse energy (compare Figure 6c and Figure 7c). The entire waveform becomes narrower at the same time. A similar effect is also observed for the GR region (Figure 6d and Figure 7d). Conversely, the behavior of the signal recorded close to p-stops seems to be the opposite.

Typically, the signal is expected to become faster with increased bias since a higher bias means a higher electric field and reduction of the plasma effect (the plasma dispersion should become faster). However, this seems to be a valid assumption only when plasma is created in a highly doped region (pad) where the electric field is high, and plasma is created even at a lower bias (since the electric field is sufficiently high to allow plasma to build at a faster rate). In a low-doped region, more time and a higher bias are needed to build up the denser plasma condition. The fact that the rising peak is shifted in time to a higher value with increased HV bias means that the plasma is created inside the IP region only at a significantly higher bias (compared to the pad).

The most striking changes in the waveform shape happen in the p-stop region at high bias. For 5 pJ pulses at 180 V, a pronounced broadening effect of the waveform occurs, as demonstrated in Figure 8a. As a result, the falling edge of the signal is drastically prolonged, indicating the possible significant temperature dependence of the plasma erosion. Further bias increase, up to 200 V, causes a massive increase in the signal amplitude (see Figure 8b). A comparison of the signals at these extreme conditions reveals that the initial rising time is comparable to the other waveforms (see Figure 8a). However, it exhibits a further increase towards an order of magnitude larger amplitude. Thus, at 200 V, an increase in amplitude for the signal recorded in the vicinity of p-stops seems to occur in two distinctive stages, with different speeds of signal growth. After reaching the first maximum, the signal increase rate changes. Then, the second maximum is reached after a very prolonged time (several nanoseconds), and the signal starts decreasing. Unfortunately, the amplifier saturation effects prevent quantitative analysis of the charge collection under these conditions (playing a role for signals above 800 mV). Obviously, with the laser power of 5 pJ, we reach the turning point for bias voltage where a slight shift in bias towards a higher value causes a dramatic increase in the signal amplitude and pulse duration. This enlarged rising time, during which the second peak reaches its maximum, might be related to the ambipolar phase, which is more dominant during the high-charge intensity injection than when the low intensity injection is applied. The second peak perhaps indicates the onset of plasma erosion, after which the signal starts to decay (bipolar phase).

It is important to note that the laser intensity was not calibrated for a PIN diode. Thus, conversion of charge to the equivalent number of minimum ionizing particles (MIPs) is not given through the text. For proper conversion of collected charge to the equivalent number of MIPs, a good knowledge of gain and correlation between laser intensity and gain suppression are required. This was out of the scope of this analysis. However, assuming that gain is suppressed due to large charge density (no charge multiplication) we estimate that around 1500 MIPs and 300 MIPs are equivalent to the collected charge at the injected laser power of 5 pJ and 0.2 pJ, respectively, in 45 microns of device.

## 4. Conclusions and Future Research Directions

In this paper, we examined the transient response of the prototype segmented LGAD sensor and discussed the role of high- and low-injection effects in high-doped and low-doped regions. The investigated sample was UFSD-like Type 10 (with two p-stops and bias grid) produced in the TI-LGAD RD50 batch for s comparison study. The presented results will be compared with an analogical study on the corresponding trench-isolated detectors.

The main finding of this work is that avalanche-like charge multiplication is observed in the IP region in the vicinity of the p-stop, where the highest electric field is expected. In addition, the recorded space–charge profiles confirmed that the complex isolation structures (two p-stops + bias ring) in the IP region can be well resolved with the proposed experimental approach. The measured IP distance agrees well with the nominal value reported by the sensor manufacturer (FBK).

During the next ELI ERIC campaign, expected to be conducted in 2023 at the ELI Beamlines facility, we plan to measure the same sample irradiated by neutrons at the TRIGA Mark Reactor at the Jozef Stefan Institute, Ljubljana, Slovenia. It is expected that the damage caused by irradiation will impact the doping level in the IP region and consequently affect the charge multiplication effects.

## Figures and Tables

**Figure 1 sensors-23-06746-f001:**
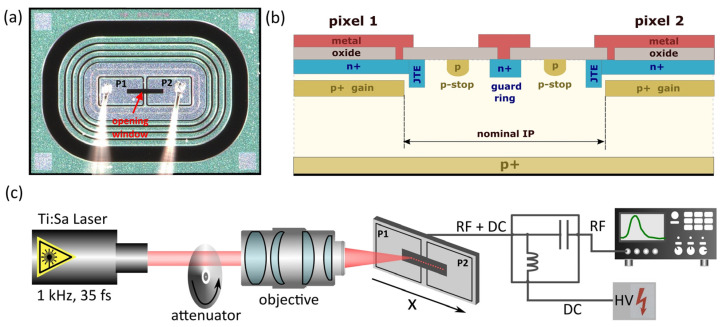
(**a**) Top view of a Type 10 LGAD prototype (2 × 1 pixel) with opening window. (**b**) Simplified visualization of the cross-section of the interpad region with two p-stops and guard ring. (**c**) Scheme of experimental configuration of TCT setup for charge-space scanning.

**Figure 2 sensors-23-06746-f002:**
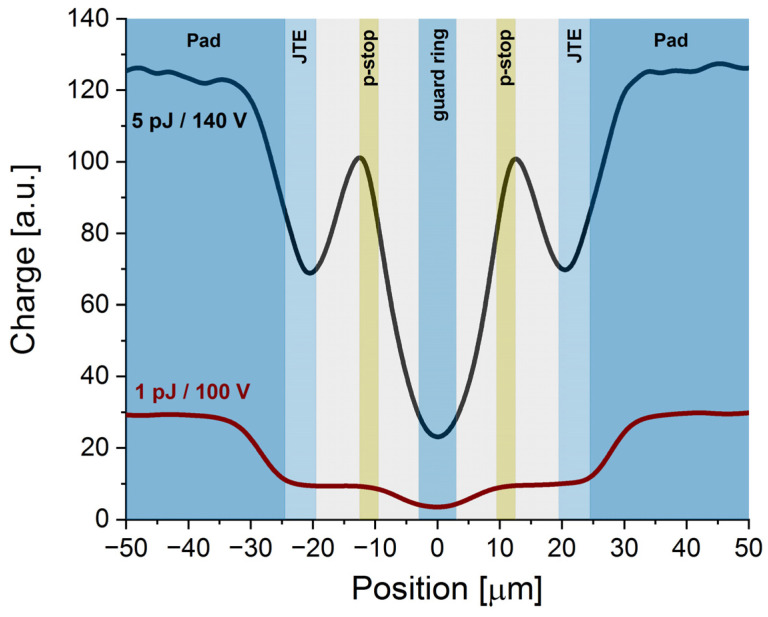
Space–charge profiles of the interpad region of the Type 10 LGAD sensor were recorded for low (1 pJ/100 V) and high (5 pJ/140 V) charge injection conditions. The background of the graph represents the estimated position and size of the corresponding elements of the interpad area.

**Figure 3 sensors-23-06746-f003:**
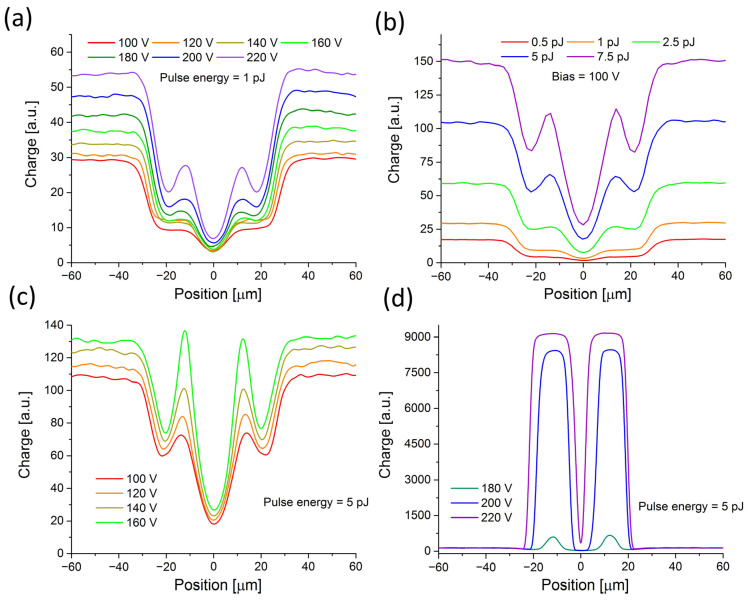
Space–charge profiles of the IP region of Type 10 LGAD sensor recorded at different charge injection conditions; (**a**) bias dependence at 1 pJ pulse energy; (**b**) pulse energy dependence at 100 V bias; (**c**) low bias dependence at 5 pJ pulse energy; (**d**) high bias dependence at 5 pJ.

**Figure 4 sensors-23-06746-f004:**
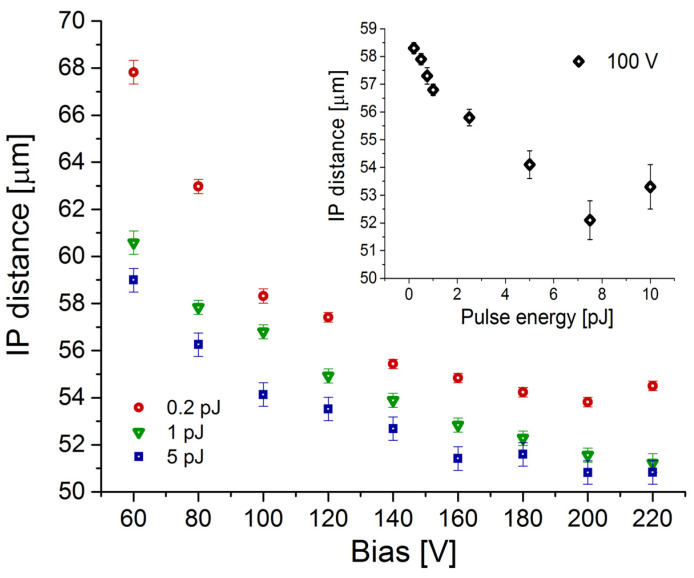
IP distance vs. bias for different laser pulse energies. Inset: IP distance vs. laser pulse energy at 100 V bias.

**Figure 5 sensors-23-06746-f005:**
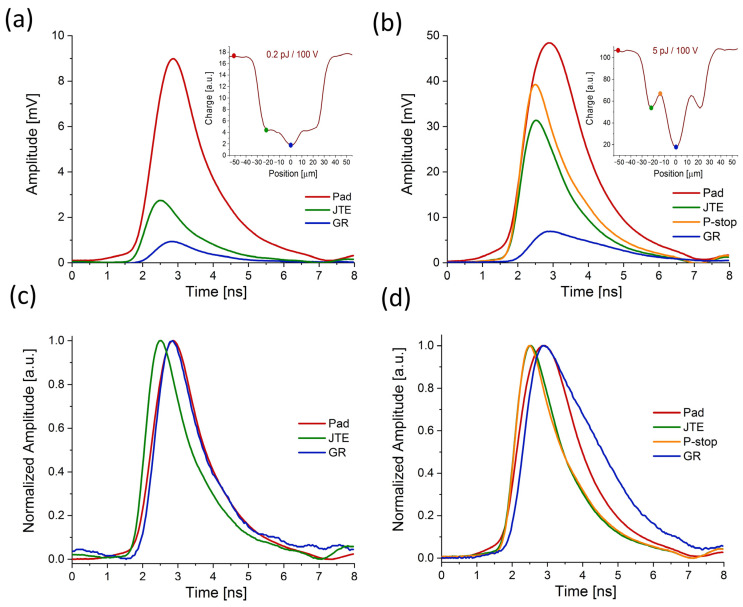
Transient current waveforms recorded at different positions (marked in the insets) of a Type 10 LGAD sensor; (**a**) waveforms recorded at 0.2 pJ and 100 V; (**b**) waveforms recorded at 5 pJ and 100 V; (**c**,**d**) the identical waveforms with normalized amplitudes.

**Figure 6 sensors-23-06746-f006:**
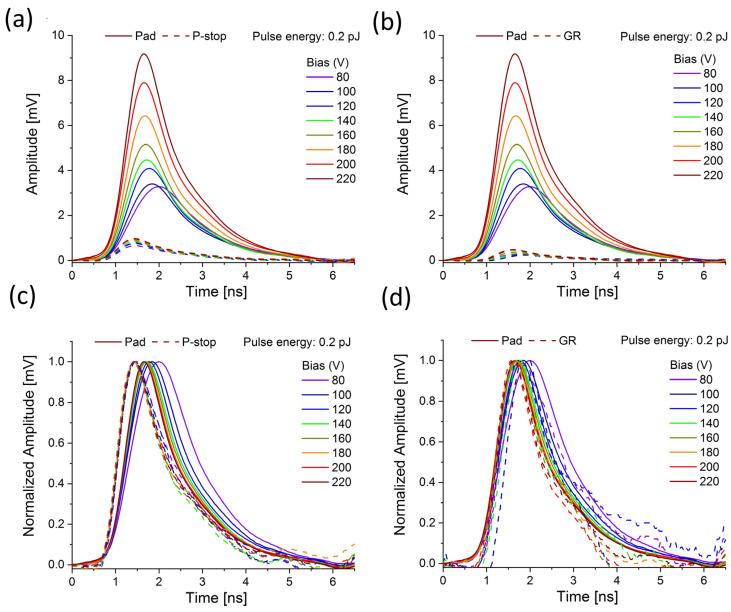
Comparison of the transient current waveforms recorded at 0.2 pJ and different positions of the Type 10 LGAD sensor; (**a**) bias scan of pad and p-stop waveforms; (**b**) bias scan of pad and GR waveforms; (**c**,**d**) the identical waveforms with normalized amplitudes.

**Figure 7 sensors-23-06746-f007:**
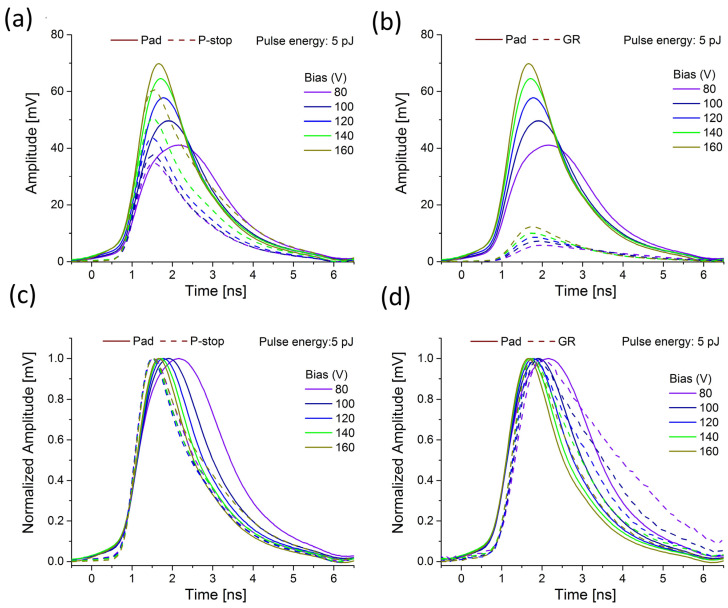
Comparison of the transient current waveforms recorded at 5 pJ and different positions of the Type 10 LGAD sensor; (**a**) bias scan of pad and p-stop waveforms; (**b**) bias scan of pad and GR waveforms; (**c**,**d**) the identical waveforms with normalized amplitudes.

**Figure 8 sensors-23-06746-f008:**
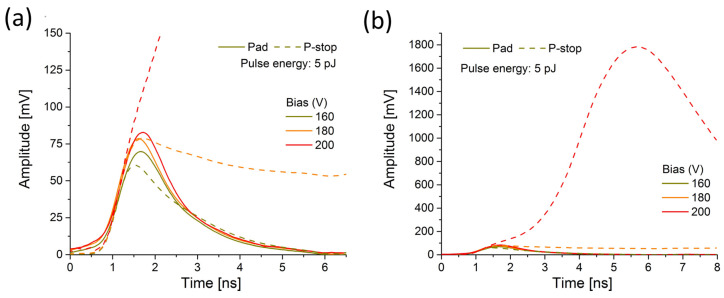
Comparison of the transient current waveforms recorded at pad and p-stop of the Type 10 LGAD sensor under very high charge injection conditions (5 pJ, 160–200 V); (**a**) comparison of the initial rising time; (**b**) comparison of the amplitudes (the same waveforms in the extended scales).

## Data Availability

The data presented in this study are available on request from the corresponding author.

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
