# Peer review of "Exploring the Interpad Gap Region in Ultra-Fast Silicon Detectors: Insights into Isolation Structure and Electric Field Effects on Charge Multiplication"

_sensors, 2023, doi:10.3390/s23156746_

Round 1
Reviewer 1 Report
Very interesting paper. The quality of some figures is low:
Figure 3: not really sharp (blurred) and letters too small
same for Figure 4 (here the letters are ok, but points are blurred), 5, 6, 7, 8
Line 52: neq should be defined
Some misprints should be improved:
Line 58 "O(10) us" --> O(10) µs
Line95 "P-stop --> p-stop
Line 157 "X-profiles" --> x-profiles (small x) same line 172
Line 168 "enrgy" --> energy
Line 169 "decrese" --> decrease
Author Response
Line 52: neq should be defined
This units is commonly used. It means: 1 MeV neutrons equivalent flux, used as a measure for displacement damage dose (DDD)

Reviewer 2 Report
In the submitted manuscript, the authors have done detailed investigations on the charge-collection behavior of the Trench-Isolated Low Gain Avalanche Diode (TI-LGAD) using a femtosecond micro-focused laser. The collected charge by the readout electrodes has been measured as a function of injection position, laser pulse energy, and bias voltage. In addition, the transient signal induced by laser injection at a few positions (below pad, JTE, p-stop, and guard ring) have been investigated in more detail and their difference in the signal are explained. The interpad (IP) distance has also been determined; the authors found a dependence of the IP distance on the bias voltage for different laser pulse energies.
Please find below a list of suggested corrections and questions which should be answered in the manuscript or in the response to the reviewer's comments.
1) L-36: 10ˆ16 cmˆ3 -> 10ˆ16 cmˆ-3
2) L-41 & 42: "shortening" -> "thinning"
3) L-50: It is mentioned that "a very uniform weighting field" is one of the key points of LGADs optimized for timing. Could you please explain how the weighting field impacts the timing? What is the consideration here? Or at least provide a proper reference here?
4) L-58: "O(10) us" -> do you mean "O(10) ps"?
5) L-72: "The shortest disctance..." -> "The thinnest substrate..."
6) L-101: "thep-stop" -> "the p-stop" with a space missing
7) Figure 1: Adding a top view of the investigated sensor will be helpful. From the cross-section, it can be seen the metal is covering the pad; there should be an opening somewhere in the design for the laser to avoid its complete absorption in the metal layer.
8) L-119 to line-124: implementing a guard ring and 2 p-stop in-between 2 pads does not seem to be a conventional design for pixellated sensor. Could you please explain why such a design is implemented and what one expects to learn from this design? In particular, the guard ring is biased to ground and thus an incomplete charge collection may occur due to charge diffusion when the e/h carriers are generated in the region close to the border of the gain layer.
9) Figure 2: At x=-50 (Pad), the charge for 1 pJ / 100 V is about 30 a.u.; for 5 pJ / 140 V is 125 a.u. The latter has a much lower output compared to the expected value (at least > 5 taking into account the ratio of the laser power and the higher bias voltage). Is it due to the plasma effect at high laser power? It would be good to have a couple of sentences here to explain it.
10) Figure 4 and line-222 to line-231: Could you explain why the IP distance depends on the laser pulse energy/intensity?
11) Figure 5(c): Why are the transient signals for "Pad" and "GR" almost the same? For the carriers generated by laser injection below GR, don't they need more time to be collected by the pad, compared to the injection below the pad?
12) The laser pulse energy of 0.2 pJ, 1 pJ, and 5 pJ has been used in the investigation. It would be good to indicate their corresponding numbers of carriers generated per micron. This may help the readers to correlate them to MIPs and understand the onset of the plasma effect.
Author Response
L-50: It is mentioned that "a very uniform weighting field" is one of the key points of LGADs optimized for timing. Could you please explain how the weighting field impacts the timing? What is the consideration here? Or at least provide a proper reference here?
Here we will explain the impact of the weighting field on the timing performance of device. The non-uniformity of weighting field brings the distortion of signal [9] and [10]. As pointed out in ref [9], in every particle detector, the shape of the induced current signal can be calculated using Ramo's theorem [10] that states that the current induced by a charge carrier is proportional to its electric charge q, the drift velocity v and the weighting field Ew: i(t) ~qvEw. This relation indicates the two key points in the design of sensors for accurate timing. First, the drift velocity needs to be constant throughout the volume of the sensor, otherwise the non-uniform drift velocities induce variations in signal shape as a function of the hit position; as results the overall time resolution is spoiled. The easiest way to obtain uniform drift velocity throughout the sensor is to have an electric field high enough to move the carriers with saturated drift velocity. Second, the sensor width should be similar to pitch and much larger than the thickness of sensor.
Ref9 (reported in text) -added: in revised manuscript.
- Cartiglia et al, Tracking in 4 dimensions, Nuclear Instruments and Methods in Physics Research Section A: Accelerators, Spectrometers, Detectors and Associated Equipment, Volume 845, 11 February 2017, Pages 47-51, https://doi.org/10.1016/j.nima.2016.05.078
Ref 10 (reported on text) – added in a revised manuscript.
- Ramo, Currents induced by electron motion, in: Proceedings of the IRE, vol. 27(9), 1939, pp. 584–585. http://dx.doi.org/10.1109/JRPROC.1939.228757
8) L-119 to line-124: implementing a guard ring and 2 p-stop in-between 2 pads does not seem to be a conventional design for pixellated sensor. Could you please explain why such a design is implemented and what one expects to learn from this design? In particular, the guard ring is biased to ground and thus an incomplete charge collection may occur due to charge diffusion when the e/h carriers are generated in the region close to the border of the gain layer.
The implementing a guard ring and p-stop structures in-between two pads is not a conventional design for pixellated sensors. Here we explain the reason. There are two ways of isolating the electrodes in p-type sensors with p-stop. Either a common p-stop (both pads share a p-stop) or individual p-stop (each pad has its own p-stop). The latter is a p-stop design that ensures highest isolation (2 p-stops). The bias ring in-between was a feature used to additionally reduce the peak fields in case of floating pads. It represents the super safe design on the expense of slightly larger IP distance.
9) Figure 2: At x=-50 (Pad), the charge for 1 pJ / 100 V is about 30 a.u.; for 5 pJ / 140 V is 125 a.u. The latter has a much lower output compared to the expected value (at least > 5 taking into account the ratio of the laser power and the higher bias voltage). Is it due to the plasma effect at high laser power? It would be good to have a couple of sentences here to explain it.
Yes, you are correct. Comparing the collected charge for 1pJ at 100 V and for 5pJ at 140 V one can conclude that the later has a much lower output compared to the expected value (as you said at least > 5 taking into account the ratio of the laser power and the higher bias voltage). The reason is so called Gain Suppression. Studies, performed within RD50 framework show that the gain of LGADs highly depends on the charge carrier density inside the gain layer. Screening of the electric field by the large density of induced charge in the gain layer can be so significant that the electrons do not undergo the impact ionisation; this reduces the total charge collection as observed in the Figure 2; The reason for reduced gain is the polarization of gain layer that in return reduces the external electric field.
10) Figure 4 and line-222 to line-231: Could you explain why the IP distance depends on the laser pulse energy/intensity?
By analysing data presented in Figure 4 we deduce that the interpad distance decreases with the increased laser power. This can be explained as follows. By increasing the laser power, the induced charge density is increased. The increased charge density as already explained leads to the gain suppression in device. However, the charge collection still increases with increased laser power but not with the rate as it should if the gain would not be suppressed (the local electric field in gain layer is screened by large charge density resulting in reduced impact ionization and thus the reduced charge multiplication). From other side, a strong excess of charge collection is observed in the intrepad region which increases with the increased laser power. All over, this leads to an artificially decreased interpad distance compared to the case when the lower laser power is used.
11) Figure 5(c): Why are the transient signals for "Pad" and "GR" almost the same? For the carriers generated by laser injection below GR, don't they need more time to be collected by the pad, compared to the injection below the pad?
Figure 5c also indicates that the transient signal for “pad” and “GR” are almost the same. For the carriers generated by laser injection below GR it is intuitively expected that they need more time to be collected, comparted to the injection below the pad. However, we see that rising time are almost the same. Slower drift velocity for carriers injected in gain is caused by plasma effect which increases their drift time.
12) The laser pulse energy of 0.2 pJ, 1 pJ, and 5 pJ has been used in the investigation. It would be good to indicate their corresponding numbers of carriers generated per micron. This may help the readers to correlate them to MIPs and understand the onset of the plasma effect.
It is important to note that the laser intensity was not calibrated for a PIN diode. Thus, conversion of charge to the equivalent number of Minimum Ionizing Particles (MIPs) is not given through the text. For proper conversion of collected charge to equivalent number of MIPs, a good knowledge of gain and correlation between laser intensity and gain suppression are required. This was out of the scope of this analysis. However, assuming that gain is suppressed by large charge density (no charge multiplication) we estimate that around 1500 MIPs and 300 MIPs are equivalent to charge collected at the injected laser power of 5 pJ and 0.2 pJ, respectively, in 45 microns of device.

Reviewer 3 Report
This work reports the study of Charge Multiplication in the Interpad Regions of a Ultra-Fast Silicon Detector. The background of USFD sensors and the practical significance is well presented. The investigation on the space-charge interpad region characterization is well performed using the transient current technique. Overall, I recommend for publication.
Here is a minor improvement for the authors to consider. In Section 2.3. it is better to supply an actual picture (or a schematic diagram) showing the experiment setup, so that readers don’t have to image what the setup look like.
Author Response
Thank you for your suggestions, which have been revised in the manuscript
Reviewer 4 Report
The paper did not provide enough information.
What kind of materials is used? What is the size of each part of the structure shown in figure 1? What are the main processes used in fabrication?
The fundamental principles of the space charge measurement should be explained basically. How the measured current is related to the space charge?
Line 36:”ρA ∼1016cm3” what is ρA ? it is difficult to understand. I guess the m3 should be cm-3, please check it.
Line 45: Noise ratio (S/N) is improved: for a 45 µm thick UFSD pixel and a gain of 20-30 a time resolution of ∼30 ps [6]… English expression should be improved.
Line 63: a p-doped implant what is the meaning of this term?
Line 345: “The main finding of this work is observed… “ please correct the grammatical mistake.
English expression should be improved.
Author Response
What kind of materials is used? What is the size of each part of the structure shown in figure 1? What are the main processes used in fabrication?
Device with layouts of 1×2 pixels of 250×375μm2 size for each pixel has been measured. In Ti-LGAD technology a two-dimensional array of small pixels implemented in the surface of a silicon die. Each pixel is an LGAD, i.e. a silicon detector with a gain implant providing a multiplication in the order of around 5 to 20. The isolation between neighbouring pixels is provided by etching the physical trenches. In reference LGAD TYPE 10, produced in Ti-LGAD batch, instead of physical trench, the isolation between pixels is provided by 2p-stops and bias ring in between them. The size of each pixel is 125x375 μm2, with an opening window of 284 μm along the x-axes of device with 1x2 pixels. More details about the size of interfaced structures in interpixel region are not publicly given by FBK. Fabrication process, and processing parameters are not published by FBK.
The fundamental principles of the space charge measurement should be explained basically. How the measured current is related to the space charge?
- The space charge within the device affects the electric field distribution. The TCT technique allows for the indirect measurement of the electric field profile by studying the movement and collection of the charge carriers. Variations in the electric field due to the presence of space charge influence the drift velocities and recombination rates of the carriers, which are reflected in the transient current measurements. To conclude, decrease/increase of the velocity during the movement of the charge is related to the changes in electric field strength, which in turn is related to the space charge.
- The TCT technique provides also insights into the carrier lifetime within the semiconductor device. The transient current profile reflects the rate at which the generated carriers are collected and recombine in the presence of the space charge. The carrier lifetime, which is the characteristic time for carrier recombination, affects the shape and duration of the transient current waveform. By analyzing the temporal behavior of the transient current, information about the carrier lifetime and the impact of the space charge on carrier
In the Experimental section it is explained how the transient current is transformed into charge:
Individual transient current waveforms were recorded at different HV biases and for different pulse energies when the beam was scanned (with 0.5 µm step) across the metallization opening window extending over two neighboring pixels (pads) and the interpad region of the investigated sensor. The space-charge profiles (x-profiles) were constructed by integrating every individual waveform.
Ref1
- Ramo, “Currents Induced by Electron Motion”, Proceedings of I.R.E. 27 (1939) p. 584.
Ref2
- Gatti, G. Padovini, V. Radeka, "Signal evaluation in multielectrode radiation detectors by means of a time dependent weighting vector”, Nucl. Instr and Meth. 193 (1982) p. 651
Line 36:”ρA ∼1016cm3” what is ρA ? it is difficult to understand. I guess the m3 should be cm-3, please check it.
We apologize. Thank you for corrections. You are right. The quantity ρA is not correctly written. Instead of ρA it should be ρA – the charge density. The values 1016 cm3 is also not correctly written; it should be 1016cm-3.
Line 63: a p-doped implant what is the meaning of this term?
One possible isolation technique of adjacent strips is the p-stop structure which is a p-type material implantation – p-doped implant.

Reviewer 5 Report
A femtosecond laser beam and the transient current technique as probing instruments are presented in this paper to investigate the interpad gap region in the Ultra Fast Silicon. The UFSD-like TYPE 10 produced in the TI-LGAD RD50 batch was used as the investigated sample. These results from experimental studies on the dynamic properties will contribute valuable insights into the IP region's isolation structure and electric field effects on charge collection.
There are also some suggestions.
1. All the pictures in this paper are not clear, and even some words in the pictures cannot be identified. The quality of the pictures needs to be improved.
2. The conclusion section contains more study ideas for the future, and it is recommended to focus on the summary of the results of this study.
3. It is recommended to weigh the title of the paper. The title does not well reflect the research significance of the paper.
Author Response
- The conclusion section contains more study ideas for the future, and it is recommended to focus on the summary of the results of this study.
We removed part that contains study ideas for future, so we follow your recommendation to stay focus on the summary of this study

Round 2
Reviewer 4 Report
The paper is improved and can be published.